# Adaptive Experimental Design with Temporal Interference: A Maximum Likelihood Approach

**Peter Glynn, Ramesh Johari, Mohammad Rasouli**
Stanford University, Stanford, CA, 94305
`{glynn, rjohari, rasoulim}@stanford.edu`

## Abstract

Suppose an online platform wants to compare a treatment and control policy, e.g., two different matching algorithms in a ridesharing system, or two different inventory management algorithms in an online retail site. Standard experimental approaches to this problem are biased (due to temporal interference between the policies), and not sample efficient. We study optimal experimental design for this setting. We view testing the two policies as the problem of estimating the steady state difference in reward between two unknown Markov chains (i.e., policies). We assume estimation of the steady state reward for each chain proceeds via non-parametric maximum likelihood, and search for consistent (i.e., asymptotically unbiased) experimental designs that are efficient (i.e., asymptotically minimum variance). Characterizing such designs is equivalent to a Markov decision problem with a minimum variance objective; such problems generally do not admit tractable solutions. Remarkably, in our setting, using a novel application of classical martingale analysis of Markov chains via Poisson's equation, we characterize efficient designs via a succinct convex optimization problem. We use this characterization to propose a consistent, efficient online experimental design that adaptively samples the two Markov chains.

## 1 Introduction

Suppose an online platform wants to compare a treatment and control policy, e.g., two different matching algorithms in a ridesharing system, or two different inventory management algorithms in an online retail site. Standard randomized controlled trials are typically not feasible, since the goal is to estimate policy performance on the entire system. Instead, the typical current practice involves dynamically alternating between the two policies for fixed lengths of time, and comparing the average performance of each over the intervals in which they were run as an estimate of the treatment effect; this is referred to as a *switchback* experimental design [4, 17].

However, switchback designs suffer from *temporal interference*: the *initial condition* in each interval of each policy is determined by the *previous* interval of the other policy, and so standard estimation techniques in this setting are *biased*. Bias due to temporal interference has been observed in ridesharing [5], in delivery services [15], and ad auctions [3]. Further, the simple, non-adaptive nature of such designs implies they are not *sample efficient*. Optimal consistent, efficient experimental design in this setting has remained a significant theoretical and practical challenge.

Our paper provides an optimal experimental design within a benchmark theoretical model for settings with temporal interference (Section 2). The central challenge posed by temporal interference is the following: we are effectively allowed only one real-world run of the system, with only finitely many observations. On the other hand, we need to use this single run to estimate performance of *both* the systems induced by each of the treatment and control policies. We model the problem by viewing each policy as its own *Markov chain* on a common underlying state space. The experimental

design problem is then to estimate the difference in the steady state reward under the treatment and control Markov chains, using only one run of the system, and without prior knowledge of any of the parameters of either policy or their rewards.

Our key contribution is a characterization of consistent and asymptotically efficient policies when estimation proceeds via maximum likelihood. In particular, we model the unknowns nonparametricaly, and use an associated nonparametric maximum likelihood estimator (MLE; Section 3). At any time step, given the current system state, the experimental design chooses which chain to sample. We restrict attention to policies that satisfy a weak regularity requirement we call *time-average regularity* (TAR; Section 4). n particular, with TAR policies, we show the MLE is consistent.

In Section 5, we present our main result: we characterize efficient TAR policies, i.e., those for which the MLE achieves asymptotically minimum variance among all TAR policies. Our approach uses a novel application of classical martingale analysis of Markov chains via Poisson's equation, and leads to a characterization of efficient designs via a succinct convex optimization problem. This simple characterization is somewhat remarkable: Markov decision problems (MDPs) with variance minimization as the objective have historically not admitted structurally simple solutions (see below). In Section 6, we use this characterization to construct an efficient, consistent adaptive online experimental design when estimation proceeds via the MLE. We conclude in Section 7.

**Related work**. *Interference* occurs in experiments whenever the outcome of a given experimental unit depends on the assignment status of *other* experimental units to either treatment or control. Recent work has devoted extensive attention to interference in experimental design for *networks* [2, 8, 12, 19, 22] and *marketplaces* [16, 20, 25, 14].

As our approach involves solving a MDP with unknown primitives, it has some model similarities with reinforcement learning, and particularly work in pure exploration in reinforcement learning [21, 6]. The main distinction in our work is that the objective for the MDP we solve is minimum variance of the MLE.

In general, the literature on MDPs with variance minimization as the objective demonstrates the principal of optimality and dynamic programming cannot be used in the classical form for average reward MDPs [23, 24, 7, 9, 13]. In particular, the optimal policy is not necessarily Markov; it can be random; and finding the optimal policy is NP-hard [18, 26]. In contrast to these prior results, our work has thus identified a remarkably tractable MDP with variance minimization as the objective.

## 2 Preliminaries

In this section we introduce the basic formal framework we employ throughout the paper. We develop the relevant notation to describe two distinct Markov chains on a common state space, as well as the design of adaptive experiments to compare the long-run average reward of these two chains.

**Notation.** As is common in analysis of finite Markov chains, we view distributions as row vectors and reward vectors as column vectors as appropriate. In addition, we use "$\xrightarrow{p}$" to denote convergence in probability, and "$\Rightarrow$" to denote weak convergence of random variables.

**Time.** We assume time is discrete, and indexed by $n = 0, 1, 2, \ldots$.

**State space**. We assume a finite state space $S$.

**Two Markov chains**. We wish to compare two different Markov chains indexed by $\ell = 1, 2$ evolving on this common state space, defined by transition matrices

$$P(\ell) = (P(\ell, x, y) : x, y \in S), \quad \ell = 1, 2. \tag{1}$$

We assume both $P(1)$ and $P(2)$ are *irreducible*.

**Auxiliary randomness**. We require two sources of randomness beyond the Markov chains themselves: one that is used to generate random rewards, and the other that is used to allow experimental designs to be randomized. Accordingly, we presume the existence of mutually independent sequences of i.i.d. uniform[0, 1] random variables $U_0, U_1, \ldots$, and $V_0, V_1, \ldots$.

**Sample space and filtration**. The sample space is $\Omega = (S \times [0, 1]^2)^\infty$, with $\omega \in \Omega$ written as $\omega = ((x_n, u_n, v_n), n \geq 0)$. For $n \geq 0$, set $X_n(\omega) = x_n$ and $U_n(\omega) = u_n, V_n(\omega) = v_n$. We define the filtration $\mathcal{G}_n = \sigma\big((X_j, U_j, V_j) : 0 \leq j \leq n\big)$ for $n \geq 0$. We also let $\mathcal{G}_\infty = \sigma\big((X_j, U_j, V_j) : j \geq 0\big)$.

**Policies (experiment designs).** A sequence of random variables $A = (A_n : n \geq 0)$ is said to be a *policy* if $A_n \in \{1, 2\}$ for $n \geq 0$, and $A$ is adapted to $(\mathcal{G}_n : n \geq 0)$. A policy is also an *experimental design*: it determines how the experimenter chooses which chain to run at each time step.[1]

Note that every policy induces a probability measure on $(\Omega, \mathcal{G}_\infty)$; this probability measure has conditional distributions defined as follows, for Borel subsets $A, B \subset [0, 1]$ and with Lebesgue measure denoted $\mu$:

$$P(X_{n+1} = x, U_{n+1} \in A, V_{n+1} \in B | \mathcal{G}_n) = P(A_n, X_n, y)\mu(A)\mu(B).$$

**Rewards.** When chain $\ell$ is in state $x$ and transitions to state $y$, a random reward is obtained, independent of the past. Formally, denote the cumulative distribution function of the reward by $F(\cdot | \ell, x, y)$. Then the reward at time $n \geq 0$ is:

$$R_n = F^{-1}(V_n | A_{n-1}, X_{n-1}, X_n). \tag{2}$$

For technical simplicity, we assume that the support of $F$ is *bounded*, i.e., that rewards are bounded in magnitude.

**Stationary distributions.** Because $S$ is finite and the two matrices $P(1)$ and $P(2)$ are irreducible, there exist unique stationary distributions $\pi(\ell) = (\pi(\ell, x) : x \in S), \ell = 1, 2$ satisfying

$$\pi(\ell) = \pi(\ell)P(\ell); \tag{3}$$
$$\pi(\ell, x) \geq 0, x \in S; \tag{4}$$
$$\sum_{x \in S} \pi(\ell, x) = 1. \tag{5}$$

**Long-run average reward.** For $\ell = 1, 2$, $x \in S$, define:

$$r(\ell, x) = \mathbb{E}\big\{ R_{n+1} | X_n = x, A_n = \ell \big\} = \sum_{y \in S} P(\ell, x, y) \int_{\mathbb{R}} z F(dz | \ell, x, y). \tag{6}$$

Now define:

$$\alpha(\ell) = \sum_{x \in S} \pi(\ell, x) r(\ell, x), \ell = 1, 2.$$

This is the stationary average reward of chain $\ell$. By the ergodic theorem for Markov chains, this is also the long-run average reward associated to chain $\ell$.

**Treatment effect.** We are interested in the difference in long run average rewards between the two chains, i.e., $\alpha = \alpha(2) - \alpha(1)$. This is the *treatment effect*.

**Estimators.** An *estimator* is a sequence of real-valued random variables $\hat{\alpha} = (\hat{\alpha}_n : n \geq 0)$ that is adapted to $(\mathcal{G}_n : n \geq 0)$.

Our goal is to design a combination of a policy $A$ and an estimator $\hat{\alpha}$ to estimate $\alpha = \alpha(2) - \alpha(1)$ *consistently* and *efficiently*, in senses that we make precise in the subsequent development.

## 3 Maximum Likelihood Estimation

In this section, we develop an approach to experiment design and estimation based on a maximum likelihood approach. Given a policy, we develop the maximum likelihood estimator (MLE) for the treatment effect $\alpha$. In particular, we take a nonparametric approach in this paper, as we make no parametric assumptions on the Markov chains being studied. Thus our approach involves maximum likelihood estimation of the transition matrices, followed by inversion to obtain an MLE for the steady state distribution.

Let $\Gamma_n(\ell, x)$ to be the number of times action $i$ at state $x$ is sampled by time $n$:

$$\Gamma_n(\ell, x) := \sum_{j=0}^{n-1} I(X_j = x, A_j = \ell), \ x \in S, \ \ell = 1, 2. \tag{7}$$

Now define:

$$\hat{P}_n(\ell, x, y) = \frac{\sum_{j=0}^{n-1} I(X_j = x, A_j = \ell, X_{j+1} = y)}{\max\{\Gamma_n(\ell, x), 1\}}; \tag{8}$$

$$\hat{P}_n(\ell) = (\hat{P}_n(\ell, x, y) : x, y \in S). \tag{9}$$

The estimators $\hat{P}(1)$ and $\hat{P}(2)$ are standard maximum likelihood estimators (MLE) for the corresponding transition matrices $P(1)$ and $P(2)$.

Define the stopping time $J = \min\{n \geq 0 : \hat{P}_n(\ell) \text{ is irreducible for } \ell = 1, 2\}$. Note that $\hat{P}_n(\ell)$ will remain irreducible for $n \geq J$, since any path with positive probability under $\hat{P}_J$ will have positive probability under $\hat{P}_n$ for all $n \geq J$. Thus for each $n \geq J$, $\hat{P}_n(\ell)$ has a unique stationary distribution $\hat{\pi}_n(\ell)$ satisfying

$$\hat{\pi}_n(\ell) = \hat{\pi}_n(\ell)\hat{P}_n(\ell); \tag{10}$$

$$\hat{\pi}_n(\ell, x) \geq 0, x \in S; \tag{11}$$

$$\sum_{x \in S} \hat{\pi}_n(\ell, x) = 1. \tag{12}$$

Note that by equivariance of the MLE, since stationary distributions are functionals of the transition matrices, each $\hat{\pi}_n(\ell)$ is also the MLE for $\pi(\ell)$.

Define

$$\hat{r}_n(\ell, x) = \frac{\sum_{j=0}^{n-1} I(X_j = x, A_j = \ell)R_{j+1}}{\max\{\Gamma_n(\ell, x), 1\}}. \tag{13}$$

The preceding is the MLE of $r(\ell, x)$ along the realized sample path.

Finally, for $n \geq J$, again by equivariance of the MLE, we conclude that the resulting nonparametric MLE $\hat{\alpha}_n$ for $\alpha$ is given by the following:

$$\hat{\alpha}_n = \hat{\pi}_n(2)\hat{r}_n(2) - \hat{\pi}_n(1)\hat{r}_n(1). \tag{14}$$

For $n < J$, $\ell = 1, 2$, and $x \in S$, we arbitrarily define $\pi_n(\ell, x) = 1/|S|$, $\hat{r}_n(\ell, x) = 0$, and $\hat{\alpha}_n = 0$.

## 4 Time Average Regular (TAR) Policies

We specialize our study to the following class of policies, that satisfy a mild regularity condition. As noted at the end of this subsection in Corollary 8, all TAR policies make $\hat{\alpha}_n$ a consistent estimator of $\alpha$.

**Definition 1** *Policy A is* time-average regular (TAR) *with (possibly random) policy limits* $\gamma = (\gamma(\ell, x) : x \in S, \ell = 1, 2)$ *if:*

$$\frac{1}{n}\Gamma_n(\ell, x) \xrightarrow{p} \gamma(\ell, x) \tag{15}$$

*as $n \to \infty$ for each $x \in S, \ell = 1, 2$.*

In the sequel we typically require that $\gamma(\ell, x) > 0$ almost surely.

**Remark 2** *Note that in Definition 1, in general the policy limits will be dependent on the initial state $X_0$. We suppress this dependence in the notation, because this dependence on initial conditions will not play a significant role. In particular, the policies we suggest for efficient experimentation will lead to deterministic policy limits, with no dependence on the initial state.*

We now characterize the structure of policy limits; in particular, we show in Proposition 4 below that policy limits almost surely lie in the set $\mathcal{K}$ defined next. The proof is available in the companion technical report [10].

**Definition 3** *Define the set $\mathcal{K}$ as follows:*

$$\mathcal{K} = \Big\{ \kappa = (\kappa(\ell, x) : x \in S, \ell = 1, 2) \text{ such that:} \tag{16}$$

$$\kappa(1, y) + \kappa(2, y) = \sum_{\ell=1}^{2} \sum_{x \in S} \kappa(\ell, x) P(\ell, x, y), \quad y \in S; \tag{17}$$

$$\sum_{\ell=1}^{2} \sum_{x \in S} \kappa(\ell, x) = 1; \tag{18}$$

$$\kappa(\ell, x) \geq 0, \quad x \in S, \ell = 1, 2 \Big\}. \tag{19}$$

**Proposition 4** *Let $A$ be a time average regular policy with policy limits $\gamma = (\gamma(\ell, x) : x \in S, \ell = 1, 2)$. Then almost surely, $\gamma \in \mathcal{K}$.*

Although straightforward, the preceding proposition encodes a surprising benefit of estimation using both chains. In particular, experimental designs in this setting can benefit from *cooperative exploration*: one chain can be used to drive the system into states for which we want samples for the *other* chain (cf. the relation (17)). In the following example, we illustrate that this possibility can yield substantial benefits in estimation variance. Indeed, in the example the variance of the MLE of the treatment effect after $n$ time steps is unboundedly lower for the optimal policy, relative to the variance of the difference in the MLEs of steady state rewards obtained by running each chain in isolation for $n$ time steps.

**Example 5 (Cooperative exploration)** *We refer to the two Markov chains depicted in Figure 1. The state space for both chains is $S = \{1, \ldots, s\}$, where $s > 1$. The red chain corresponds to $\ell = 1$ and the blue chain corresponds to $\ell = 2$. The transition probabilities are as depicted in the figure. In particular, we assume that chain 1 has $P(x, x+1) = P(s, 1) = 1$ for $x = 1, \ldots, s-1$, and chain 2 has $P(x, x-1) = P(1, s) = 1$ for $x = 2, \ldots, s$.*

*We assume the experimenter* knows *the transition matrices exactly (as they are deterministic), and thus the only uncertainty in estimating the reward distribution comes from uncertainty regarding the reward distribution of each chain. We assume each chain* only *earns a reward in state $x = 1$. In particular, chain $\ell$ earns a reward that is Bernoulli($q(\ell)$) in state 1, for some unknown parameter $q(\ell)$ with $0 < q(\ell) < 1$. Clearly the stationary distribution of each chain is $\pi(\ell, x) = 1/s$, and so the steady state mean reward of each chain is $\alpha(\ell) = q(\ell)/s$. Thus the treatment effect is $(q(2) - q(1))/s$.*

*First, suppose that for $\ell = 1, 2$ we wanted to estimate only $\alpha(\ell)$ by running chain $\ell$, i.e., $A_n = \ell$ for all $n$. Then note that in every $S$ steps, only one observation is received of the reward in state 1. Let $\hat{\alpha}_n(\ell)$ denote the maximum likelihood estimate of steady state reward obtained from the first $n$ steps. Given the structure of this chain, it is straightforward to check that the MLE at time $n > s$ reduces to the sample average of $\lfloor n/s \rfloor$ independent Bernoulli($q(\ell)$) samples. This estimator has variance that scales as $\Theta(s/n)$. Thus, any attempt at estimation of the variance of steady state reward by running each chain in isolation will have variance that scales with $s$.*

*On the other hand, now suppose we use the following sampling policy: the policy always samples chain 1 when in state $s$; the policy always samples chain 2 in states $2, \ldots, s-1$; and in successive visits to state 1, the policy deterministically alternates between sampling chains 1 and 2. Suppose for simplicity that this chain starts at $X_0 = 1$. Then in every four periods, this chain obtains one independent sample each of a reward from chain 1 in state 1 (i.e., Bernoulli($q(1)$), and a reward from chain 2 in state 1 (i.e., Bernoulli($q(2)$). Thus the maximum likelihood estimator of $\alpha(\ell)$ will have variance that scales as $\Theta(4/n)$, and in particular, does* not *grow with $s$. In particular, the improvement in variance under this policy relative to the preceding approach can be made unboundedly large by increasing $s$.*

*This example illustrates the surprising insight that by* cooperatively exploring *using* both *chains together, substantial benefits in estimation variance can be achieved relative to the variance of*

*estimation with each chain in isolation. In this example, both approaches to estimation will be consistent. However, the state-dependent sampling policy leads to a substantial reduction in variance, because it benefits from cooperative exploration: for each chain $\ell = 1, 2$, the* other *chain is used to drive the system back to where samples are most needed to reduce variance. By contrast, running each chain in isolation forces the experimenter to wait $s$ time steps between successive observations of the random reward in state $1$. When $s$ becomes larger, the long run average time spent in state 1 approaches $1/2$ for the state-dependent sampling policy, but approaches zero for either chain in isolation.*

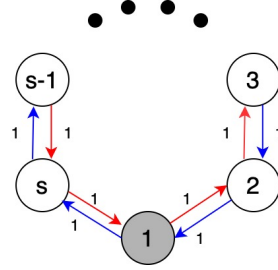

Figure 1: The two Markov chains described in Example 5. Chain 1 is red, and chain 2 is blue. Rewards are only earned in state 1 for each chain; in particular, the reward distribution in state 1 is Bernoulli($q(\ell)$) for chain $\ell$.

**Remark 6** *Given a TAR policy A, for any initial state, the law of the resulting policy limits $\gamma$ is a probability measure over $\mathcal{K}$, according to Proposition 4.*

*Conversely, suppose that $\kappa \in \mathcal{K}$ is positive (i.e. $\kappa(\ell, x) > 0$ for $x \in S, \ell = 1, 2$). We show that regardless of the initial state, $\kappa$ can be achieved as the (deterministic) policy limit of some TAR policy. For example, define:*

$$p(\ell, x) = \frac{\kappa(\ell, x)}{\kappa(1, x) + \kappa(2, x)} \tag{20}$$

*for $\ell = 1, 2$ and $x \in S$. Define $A_n$ to be the following* stationary Markov policy*:*

$$P(A_n = \ell | \mathcal{G}_{n-1}, X_n) = p(\ell, X_n) \tag{21}$$

*for $\ell = 1, 2$, $n \geq 0$. This policy is Markov because it depends only on the current state $X_n$ (and the auxiliary randomness $U_n$) and stationary because the choice probabilities do not change with time. Further, it is straightforward to check that since each $P(\ell)$ is irreducible for $\ell = 1, 2$, this policy makes $X_n$ an irreducible Markov chain. As a result, this chain therefore has a unique stationary distribution regardless of the initial state. Elementary computation yields that the stationary distribution must be $\pi(x) = \kappa(0, x) + \kappa(1, x), x \in S$. Therefore, the policy limit of this policy is equal to $\kappa$, regardless of the initial state.*

The following proposition implies Corollary 8: the MLE estimator $\hat{\alpha}_n$ is *consistent* under all TAR policies with positive policy limits, i.e., $\hat{\alpha}_n$ converges in probability to $\alpha$. Both proofs are relatively straightforward and available in the companion technical report [10]

**Proposition 7** *If A is a TAR policy with policy limits that are almost surely positive, then for $\ell = 1, 2$, as $n \to \infty$, there holds*

$$\hat{P}_n(\ell) \xrightarrow{p} P(\ell) \tag{22}$$

*and*

$$\hat{\pi}_n(\ell) \xrightarrow{p} \pi(\ell). \tag{23}$$

**Corollary 8** *If A is a TAR policy with policy limits that are almost surely positive, then the MLE estimator $\hat{\alpha}_n$ is* consistent *under A, i.e., $\hat{\alpha}_n \xrightarrow{p} \alpha$ as $n \to \infty$.*

# 5 The MLE with TAR Policies: A Characterization of Efficiency

In this section, we study the asymptotic variance of the MLE when TAR policies are used to sample and compare the two Markov chains in the experiment. In Section 5.1, we develop a central limit theorem for the MLE estimator when used with TAR policies. In Section 5.2, we use this central limit theorem to give a characterization of policies that are efficient, in the sense that they provide minimum asymptotic variance.

## 5.1 A Central Limit Theorem

A key tool in our analysis is *Poisson's equation* from the theory of Markov chains. Let $P$ be the transition matrix of an irreducible Markov chain on $S$, with corresponding stationary distribution $\pi$; let $\Pi$ be matrix with rows equal to $\pi$, i.e., $\Pi = e\pi$ where $e = (1, \ldots, 1)$. Further let $r$ be a reward function on $S$; we center $r$ by defining $\tilde{r} = r - e\pi r$. Recall that Poisson's equation for $\tilde{r}$ under $P$ is:

$$(I - P)g = \tilde{r}. \tag{24}$$

One solution to the previous equation is given by:

$$\tilde{g} = (I - P + \Pi)^{-1}r, \tag{25}$$

where $(I - P + \Pi)^{-1}$ is the *fundamental matrix* associated to $P$. (In general, the solution to Poisson's equation is not unique; however, the preceding solution is the unique one for which $\pi\tilde{g} = \pi r$.) The following result is a well-known central limit theorem for finite Markov chains (see, e.g., [1], Theorem 7.2).

**Proposition 9** *As $n \to \infty$, the random variable $\frac{1}{\sqrt{n}}\left(r(X_0) + \cdots + r(X_{n-1}) - n\pi r\right)$ converges weakly to a normal random variable with mean zero and variance $\sigma^2(r) = \pi\tilde{g}^2 - \pi(P\tilde{g})^2$, where $\tilde{g}$ is the solution to Poisson's equation in* (25).[2]

Our goal is to obtain a central limit theorem for TAR policies. Note that there are several complexities in our setting that make this challenging: in general TAR policies allow for adaptive sampling, i.e., the chain chosen by the policy at a given time step can depend on the past history. In particular, the induced state process may no longer be Markovian as a result. To further complicate matters, the MLE $\hat{\alpha}_n$ cannot be represented simply as an average sum of rewards over the first $n$ time periods.

Nevertheless, we now present a central limit theorem result analogous to Proposition 9 for MLE estimation with TAR policies. For $\ell = 1, 2$, we define $\tilde{g}(\ell)$ to be the solution to (25) for the transition matrix $P(\ell)$ with reward function $r(\ell)$, i.e.:

$$\tilde{g}(\ell) = (I - P(\ell) + \Pi(\ell))^{-1}r(\ell).$$

In addition, define:

$$\sigma^2(\ell, x) = \text{Var}\left(\tilde{g}(\ell, X_1) + R_1 \big| X_0 = x, A_0 = \ell\right) \tag{26}$$

$$= \sum_{y \in S} P(\ell, x, y)[\tilde{g}(\ell, y) - \sum_{z \in S} P(\ell, x, z)\tilde{g}(\ell, z)]^2 \tag{27}$$

$$+ \sum_{y \in S} P(\ell, x, y) \text{Var}(R_1 | X_0 = x, X_1 = y, A_0 = \ell). \tag{28}$$

We have the following theorem.

**Theorem 10** *Suppose that $A$ is a TAR policy, with policy limits $\gamma = (\gamma(\ell, x) : \ell = 1, 2, x \in S)$ that are almost surely positive. Let $G = \big(G(\ell, x) : x \in S, \ell = 1, 2\big)$ be a family of independent Gaussian random variables with mean $0$ and unit variance. Then for the MLE $\hat{\alpha}_n$, there holds*

$$n^{1/2}(\hat{\alpha}_n - \alpha) \Rightarrow \sum_{x \in S} \frac{\pi(1, x)\sigma(1, x)}{\gamma(1, x)^{1/2}}G(1, x) - \sum_{x \in S} \frac{\pi(2, x)\sigma(2, x)}{\gamma(2, x)^{1/2}}G(2, x) \tag{29}$$

*as $n \to \infty$, where $G$ is independent of $\gamma$.*

The full proof of Theorem 10 is available in [10]. A key idea in the proof is to show, via Poisson's equation, that:

$$\hat{\alpha}_n(\ell) - \alpha(\ell) = \big(\hat{\pi}_n(\ell) - \pi(\ell)\big)r(\ell) = \hat{\pi}_n(\ell)\big(\hat{P}_n(\ell) - P(\ell)\big)\tilde{g}(\ell).$$

We are then able to apply martingale arguments to analyze the right hand side of the preceding expression, by looking at the difference in the realized state transition, and the expected state transition. These steps allow us to leverage classical martingale central limit theorem results (see [10] for details).

For later reference, the following two results will be useful. The first applies Theorem 10 to show that we can lower bound the scaled asymptotic variance of the MLE. The second shows that for TAR policies that have constant and positive policy limits, in fact we can exactly obtain the scaled asymptotic variance of the MLE. We later use these results to show the existence of an optimal TAR policy with constant policy limits. Proofs of both results are in [10].

**Corollary 11** *Let $A$ be a TAR policy with almost surely positive policy limits $\gamma$, with associated MLE $\hat{\alpha}_n$. Then there holds:*

$$\liminf_{n\to\infty} n\,\mathrm{Var}(\hat{\alpha}_n - \alpha) \geq \sum_{\ell=1,2}\sum_{x\in S}\frac{\pi^2(\ell,x)\sigma^2(\ell,x)}{E\{\gamma(\ell,x)\}}. \tag{30}$$

**Corollary 12** *Let $A$ be a TAR policy with almost surely constant and positive policy limits $\gamma$. Let $\hat{\alpha}_n$ be the associated MLE. Then:*

$$\lim_{n\to\infty} n\,\mathrm{Var}(\hat{\alpha}_n - \alpha) = \sum_{\ell=1,2}\sum_{x\in S}\frac{\pi^2(\ell,x)\sigma^2(\ell,x)}{\gamma(\ell,x)}. \tag{31}$$

## 5.2 Optimal TAR Policies with the MLE Estimator

In this section we characterize efficient policies, i.e., those that achieve minimum asymptotic variance within the class of TAR policies with almost surely positive policy limits. (Note that we know such policies are consistent from Corollary 8, and thus we focus solely on asymptotic variance in considering efficiency.) We show that the policy limits of any such policy can be characterized via the solution to a particular convex optimization problem. We require the mild additional assumption in this section that $\sigma(\ell,x) > 0$ for all $\ell, x$; this will hold, e.g., if rewards are random in each state. We start with the following formal definition of efficiency.

**Definition 13** *Let $A^*$ be a TAR policy with almost surely positive policy limits, with associated MLE $\hat{\alpha}_n^*$. We say that $A^*$ is* efficient *if it has lower (scaled) asymptotic variance than any other TAR policy with almost surely positive policy limits; i.e., for any such policy $A$ with associated MLE $\hat{\alpha}_n$, there holds:*

$$\limsup_{n\to\infty} n\,\mathrm{Var}(\hat{\alpha}_n^* - \alpha) \leq \liminf_{n\to\infty} n\,\mathrm{Var}(\hat{\alpha}_n - \alpha). \tag{32}$$

The following theorem leverages the central limit theorem in Theorem 10, and in particular Corollaries 11 and 12, to give an optimization problem whose solution characterizes the optimal (scaled) asymptotic variance. The proof of the theorem is in [10].

**Theorem 14** *Suppose that for all $\ell, x$, there holds $\sigma(\ell,x) > 0$. Consider the following (convex) optimization problem:*

$$minimize \quad \sum_{\ell=1}^{2}\sum_{x\in S}\frac{\pi^2(\ell,x)\sigma^2(\ell,x)}{\kappa(\ell,x)} \tag{33}$$

$$subject\ to \quad \kappa \in \mathcal{K}. \tag{34}$$

*This problem has a unique solution $\kappa^*$, and all entries of $\kappa^*$ are positive. Any TAR policy $A^*$ that has almost surely constant policy limits $\kappa^*$ is efficient, and the scaled asymptotic variance of the MLE under $A^*$ is given by (33) evaluated at $\kappa^*$; i.e., if we let $\hat{\alpha}_n^*$ denote the resulting MLE, we have:*

$$\lim_{n\to\infty} n\,\mathrm{Var}(\hat{\alpha}_n^* - \alpha) = \sum_{\ell=1}^{2}\sum_{x\in S}\frac{\pi^2(\ell,x)\sigma^2(\ell,x)}{\kappa^*(\ell,x)}. \tag{35}$$

.

# 6 An Online Experimental Design: OnlineETI

Based on Theorem 14, it follows that one efficient policy is the stationary Markov policy obtained by inserting $\kappa^*$ in (20). However, such a policy requires knowledge of the system parameters (as these are required to solve the optimization problem (33)-(34) that yields $\kappa^*$). Of course, if these parameters were already known, there would be no need for experiment design and estimation in the first place.

In this section, we instead construct an *online* policy (i.e., one that does not use *a priori* knowledge of system parameters) that is consistent and efficient in the limit as $n \to \infty$. In particular, the policy we construct will be TAR with policy limits $\kappa^*$. Our proposed policy, called OnlineETI (for *Online Experimentation with Temporal Interference*) works as follows. At every time step $n$, OnlineETI maintains an MLE of $P(\ell)$ as $\hat{P}_n(\ell)$. Initially, in every state, the policy samples chain $\ell = 1, 2$ with probability 0.5; this continues until $\hat{P}_n(\ell)$ becomes irreducible (with the associated stationary distribution denoted $\hat{\pi}_n(\ell)$). OnlineETI estimates the mean and variance of one-step rewards as well, for each triple $(A_n, X_n, X_{n+1}) = (\ell, x, y), \ell = 1, 2, x, y \in S$. These estimates are used to estimate $\tilde{g}(\ell)$, and thus yield an estimate $\hat{\sigma}_n(\ell, x)$ for $\sigma(\ell, x)$ (cf. (28)).

Using these estimates, OnlineETI computes $\hat{\kappa}_n$ as the minimizer of $\sum_{\ell=1,2} \sum_{x \in S} \hat{\pi}_n(\ell, x)^2 \hat{\sigma}_n(\ell, x)^2 / \kappa(\ell, x)$ over $\kappa \in \mathcal{K}$ (cf. (33)-(34)). At each time step $n$, OnlineETI then samples from chain $\ell$ with the following probability:

$$\hat{p}_n(\ell, x) = \left(1 - M_n(x)^{-1/2}\right) \left(\frac{\hat{\kappa}_n(\ell, x)}{\hat{\kappa}_n(1, x) + \hat{\kappa}_n(2, x)}\right) + \frac{1}{2} M_n(x)^{-1/2}, \tag{36}$$

where $M_n(x)$ is the cumulative number of visits to state $x$ up to time $n$. In other words, $\hat{p}_n(\ell, x)$ is chosen proportional to $\hat{\kappa}_n(\ell, x)$, but with an additional $1/2 \cdot M_n(x)^{-1/2}$ probability of playing either chain. This latter term is *forced exploration*: it ensures sufficient exploration of both chains to give consistent estimates of model parameters, without influencing the policy limits, and thus the asymptotic variance, of the policy. As $n \to \infty$, we show that $\hat{\kappa}_n(\ell, x) \to \kappa^*(\ell, x)$ almost surely, and thus that $\hat{p}_n(\ell, x)$ converges to the choice probability of the optimal stationary Markov policy defined by inserting $\kappa^*$ in (20). The full pseudocode of OnlineETI is in [10]. We have the following theorem; the full proof also appears in [10].

**Theorem 15** *Suppose that for all $\ell, x$, there holds $\sigma(\ell, x) > 0$. Then* OnlineETI *is a TAR policy with policy limits $\kappa^*$ (as defined in Theorem 14), and thus it is consistent and efficient.*

# 7 Conclusion

In this paper we focus on asymptotic efficiency rather than fixed finite horizons. We note that in fact, the term $M_n(x)^{-1/2}$ in (36) can be replaced by $M_n(x)^{-\beta}$ for any $0 < \beta < 1$, and the result of Theorem 15 still holds. We conjecture that $\beta$ serves as a tuning parameter between finite horizon bias and variance: informally, as $\beta$ increases, forced exploration decreases, yielding higher bias but lower variance. A formal investigation of optimal finite horizon experimental design remains an important direction for future work.

We also note that our approach is fully nonparametric, with no structural assumptions on the system. In practice, full knowledge of the state space may be infeasible. Nevertheless, we believe our main insights regarding experimental design that adaptively samples the two chains, with the goal of minimizing variance of the treatment effect estimate, can still provide guidance on optimal design in such applied domains. We have begun investigating the design of simpler experimental designs that employ estimation based on sample averages rather than maximum likelihood, with less required knowledge of the state space. This remains a promising avenue of investigation for future work.

## Broader Impact

This is a theoretical work on experimental design, and does not present any foreseeable societal consequence.

## Acknowledgments, Funding Disclosure, and Competing Interest Statement

The authors gratefully acknowledge helpful feedback from participants at the Workshop on Machine Learning and User Decision-Making, and at the Simons Institute. This work was supported by the National Science Foundation under grants 1931696 and 1839229. In addition, Ramesh Johari was a technical advisor on experimental design for Uber Technologies, Inc., for a part of the period in which this work was completed.

## Footnotes

[1]In general, the experimenter may commit in advance to a time horizon $N$ of interest, and the experimenter may use this knowledge in design of their policy. In what we study here, for ease of presentation, we presume that the policy is defined for all $n \geq 0$. In the fixed horizon setting, our results can be extended in a straightforward manner to characterize consistency and variance as $N \to \infty$ under appropriate regularity conditions.

[2]Here we use the notation $f^2$ to denote elementwise squaring of the function, i.e., $f^2(x) = f(x)^2$.

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
