[Supplementary Material]

**Supplementary Material for "Adaptive Experimental Design with**
**Temporal Interference: A Maximum Likelihood Approach"**

## A    An Example: Cooperative Exploration

Throughout this section, we refer to the two Markov chains depicted in Figure 1. The state space
for both chains is $S = \{1, \ldots, s\}$, where $s > 1$. The red chain corresponds to $\ell = 1$ and the blue
chain corresponds to $\ell = 2$. The transition probabilities are as depicted in the figure. In particular,
we assume that chain 1 has $P(x, x + 1) = P(s, 1) = 1$ for $x = 1, \ldots, s - 1$, and chain 2 has
$P(x, x - 1) = P(1, s) = 1$ for $x = 2, \ldots, s$.

We assume the experimenter *knows* the transition matrices exactly (as they are deterministic), and
thus the only uncertainty in estimating the reward distribution comes from uncertainty regarding
the reward distribution of each chain. We assume each chain *only* earns a reward in state $x = 1$. In
particular, chain $\ell$ earns a reward that is Bernoulli($q(\ell)$) in state 1, for some unknown parameter $q(\ell)$
with $0 < q(\ell) < 1$. Clearly the stationary distribution of each chain is $\pi(\ell, x) = 1/s$, and so the
steady state mean reward of each chain is $\alpha(\ell) = q(\ell)/s$. Thus the treatment effect is $(q(2) - q(1))/s$.

First, suppose that for $\ell = 1, 2$ we wanted to estimate only $\alpha(\ell)$ by running chain $\ell$, i.e., $A_n = \ell$ for
all $n$. Then note that in every $S$ steps, only one observation is received of the reward in state 1. Let
$\hat{\alpha}_n(\ell)$ denote the maximum likelihood estimate of steady state reward obtained from the first $n$ steps.
Given the structure of this chain, it is straightforward to check that the MLE at time $n > s$ reduces to
the sample average of $\lfloor n/s \rfloor$ independent Bernoulli($q(\ell)$) samples. This estimator has variance that
scales as $\Theta(s/n)$. Thus, any attempt at estimation of the variance of steady state reward by running
each chain in isolation will have variance that scales with $s$.

On the other hand, now suppose we use the following sampling policy: the policy always samples
chain 1 when in state $s$; the policy always samples chain 2 in states $2, \ldots, s - 1$; and in successive
visits to state 1, the policy deterministically alternates between sampling chains 1 and 2. Suppose
for simplicity that this chain starts at $X_0 = 1$. Then in every four periods, this chain obtains one
independent sample each of a reward from chain 1 in state 1 (i.e., Bernoulli($q(1)$), and a reward from
chain 2 in state 1 (i.e., Bernoulli($q(2)$). Thus the maximum likelihood estimator of $\alpha(\ell)$ will have
variance that scales as $\Theta(4/n)$, and in particular, does *not* grow with $s$. In particular, the improvement
in variance under this policy relative to the preceding approach can be made unboundedly large by
increasing $s$.

This example illustrates the surprising insight that by *cooperatively exploring* using *both* chains
together, substantial benefits in estimation variance can be achieved relative to the variance of
estimation with each chain in isolation. In this example, both approaches to estimation will be
consistent. However, the state-dependent sampling policy leads to a substantial reduction in variance,
because it benefits from cooperative exploration: for each chain $\ell = 1, 2$, the *other* chain is used to
drive the system back to where samples are most needed to reduce variance. By contrast, running
each chain in isolation forces the experimenter to wait $s$ time steps between successive observations
of the random reward in state 1. When $s$ becomes larger, the long run average time spent in state
1 approaches $1/2$ for the state-dependent sampling policy, but approaches zero for either chain in
isolation.

Figure 1: The two Markov chains described in Appendix A. Chain 1 is red, and chain 2 is blue.
Rewards are only earned in state 1 for each chain; in particular, the reward distribution in state 1 is
Bernoulli($q(\ell)$) for chain $\ell$.

# B Proofs: Section 4

**Proof of Proposition 4.** Relations (18) and (19) are obvious. As for (17), note that

$$\frac{1}{n}\Gamma_n(1,y) + \frac{1}{n}\Gamma_n(2,y) \tag{37}$$

$$=\frac{1}{n}\sum_{j=0}^{n-1} I(X_j = y) \tag{38}$$

$$=\frac{1}{n}\sum_{j=1}^{n} I(X_j = y) + \frac{1}{n}\big(I(X_0 = y) - I(X_n = y)\big) \tag{39}$$

$$=\frac{1}{n}\sum_{j=1}^{n} I(X_j = y) + O_p\left(\frac{1}{n}\right) \tag{40}$$

$$=\sum_{\ell=1}^{2}\sum_{x\in S}\frac{1}{n}\sum_{j=1}^{n} I(X_{j-1} = x, A_{j-1} = \ell, X_j = y) + O_p\left(\frac{1}{n}\right) \tag{41}$$

$$=\sum_{\ell=1}^{2}\sum_{x\in S}\frac{1}{n}\Gamma_n(\ell,x)P(\ell,x,y) \tag{42}$$

$$+\sum_{\ell=1}^{2}\sum_{x\in S}\left\{\frac{1}{n}\sum_{j=1}^{n} I(X_{j-1} = x, A_{j-1} = \ell)[I(X_j = y) - P(\ell,x,y)]\right\} + O_p\left(\frac{1}{n}\right). \tag{43}$$

(Here we use the notation that $f(n) = O_p(1/n)$ to denote stochastic boundedness of $nf(n)$: for all $\epsilon > 0$, there exists deterministic $M$ such that $P(|nf(n)| > M) < \epsilon$ for all $n$.)

Let $W_j = I(X_{j-1} = x, A_{j-1} = \ell)[I(X_j = y) - P(\ell,x,y)]$. This is a martingale difference sequence adapted to $\mathcal{G}_j$. In particular, as a result the $W_j$ are *orthogonal* in the sense that for $j < k$, there $E\{_jW_k\} = 0$. (This result follows by conditioning on $\mathcal{G}_j$ and nesting conditional expectations: $E\{W_jW_k\} = E\{E\{W_k|\mathcal{G}_j\}W_j\} = 0$.) Using orthogonality of the martingale differences implies that

$$E\left\{\left(\frac{1}{n}\sum_{j=1}^{n} I(X_{j-1} = x, A_{j-1} = \ell)[I(X_j = y) - P(\ell,x,y)]\right)^2\right\} \tag{44}$$

$$=E\left\{\frac{1}{n^2}\Gamma_n(\ell,x)\big(P(\ell,x,y) - P^2(\ell,x,y)\big)\right\} \to 0 \tag{45}$$

as $n \to \infty$. Therefore, by Chebyshev's inequality

$$\frac{1}{n}\sum_{j=1}^{n} I(X_{j-1} = x, A_{j-1} = \ell)[I(X_j = y) - P(\ell,x,y)] \xrightarrow{p} 0 \tag{46}$$

as $n \to \infty$. Taking the limits in (43) yields (17). ∎

**Proof of Proposition 6.** We start by proving (22). We recall that

$$\hat{P}_n(\ell,x,y) = \frac{\sum_{j=0}^{n-1} I(X_j = x, A_j = \ell, X_{j+1} = y)}{\max\{\Gamma_n(\ell,x),1\}} \tag{47}$$

As in the proof of Proposition 4,

$$\frac{1}{n}\sum_{j=0}^{n-1} I(X_j = x, A_j = \ell, X_{j+1} = y) \tag{48}$$

$$=\left\{\frac{1}{n}\sum_{j=0}^{n-1} I(X_j = x, A_j = \ell)[I(X_{j+1} = y) - P(\ell,x,y)]\right\} + \frac{1}{n}\Gamma_n(\ell,x)P(\ell,x,y) \tag{49}$$

Therefore,

$$\hat{P}_n(\ell, x, y) = \frac{\left\{\frac{1}{n}\sum_{j=0}^{n-1} I(X_j = x, A_j = \ell)[I(X_{j+1} = y) - P(\ell, x, y)]\right\}}{\frac{1}{n}\max\{\Gamma_n(\ell, x), 1\}} \tag{50}$$

$$+ \frac{\Gamma_n(\ell, x)}{\max\{\Gamma_n(\ell, x), 1\}} P(\ell, x, y) \tag{51}$$

$$= \frac{o_p(1)}{\frac{1}{n}\max\{\Gamma_n(\ell, x), 1\}} + \frac{\Gamma_n(\ell, x)}{\max\{\Gamma_n(\ell, x), 1\}} P(\ell, x, y) \quad \text{from (46)} \tag{52}$$

$$\xrightarrow{p} P(\ell, x, y) \tag{53}$$

as $n \to \infty$, where the convergence in (53) holds because $\gamma(\ell, x)$ are almost surely positive.

We now prove (23). Let $\mu_n$ denote the law of $\hat{\pi}_n$, and view it as a probability measure on vectors in the probability simplex on the state space $S$, denoted $\Delta(S)$. The set $\Delta(S)$ is compact, and so by Prohorov's theorem there exists a deterministic subsequence $n_k$ such that $\mu_{n_k}$ converges weakly to a probability measure $\mu$ on $\Delta(S)$, with associated random variable $\pi'(\ell)$. Since $\hat{P}_n(\ell) \xrightarrow{p} P(\ell)$ by (53), and $P(\ell)$ is deterministic, it follows by Slutsky's theorem that:

$$\hat{\pi}_{n_k}(\ell)\hat{P}_{n_k}(\ell) \Rightarrow \pi'(\ell)P(\ell).$$

Since the policy limits are almost surely positive, $J$ is almost surely finite. Thus, for all sufficiently large $k$ there holds $\hat{\pi}_{n_k}(\ell)\hat{P}_{n_k}(\ell) = \hat{\pi}_{n_k}(\ell)$. It follows that $\pi'(\ell) = \pi'(\ell)P(\ell)$, so that $\pi'(\ell) = \pi(\ell)$ almost surely. In other words, the measure $\mu$ is the Dirac measure that places probability one on $\pi(\ell)$. Since this is the case for every convergent subsequence of $\{\mu_n\}$, we conclude that $\hat{\pi}_n(\ell) \Rightarrow \pi(\ell)$. Since $\pi(\ell)$ is deterministic, we conclude that $\pi_n(\ell) \xrightarrow{p} \pi(\ell)$ as $n \to \infty$, as required. ∎

**Proof of Corollary 7.** Since the policy limits of $A$ are almost surely positive, it is straightforward to show that for each $\ell, x, \hat{r}_n(\ell, x) \xrightarrow{p} r(\ell, x)$ as $n \to \infty$. The result then follows from Proposition 6. ∎

## C  Proofs: Section 5

**Proof of Theorem 9.** We begin by showing that for $\ell = 1, 2$ and $n \geq J$, there holds:

$$\big(\hat{\pi}_n(\ell) - \pi(\ell)\big)r(\ell) = \hat{\pi}_n(\ell)\big(\hat{P}_n(\ell) - P(\ell)\big)\tilde{g}(\ell). \tag{54}$$

To see this, observe that for $n \geq J$,

$$\hat{\pi}_n(\ell) - \pi(\ell) = \hat{\pi}_n\hat{P}_n(\ell) - \hat{\pi}_n(\ell)P(\ell) + \hat{\pi}_n(\ell)P(\ell) - \pi(\ell)P(\ell) \tag{55}$$

so rearranging the terms we get,

$$\big(\hat{\pi}_n(\ell) - \pi(\ell)\big)\big(I - P(\ell)\big) = \hat{\pi}_n(\ell)\big(\hat{P}_n(\ell) - P(\ell)\big). \tag{56}$$

Because $\Pi(\ell)$ has identical elements down each column,

$$\big(\hat{\pi}_n(\ell) - \pi(\ell)\big)\Pi(\ell) = 0, \tag{57}$$

and hence

$$\big(\hat{\pi}_n(\ell) - \pi(\ell)\big)\big(I - P(\ell) + \Pi(\ell)\big) = \hat{\pi}_n(\ell)\big(\hat{P}_n(\ell) - P(\ell)\big). \tag{58}$$

Recall that we defined $\tilde{g}(\ell) = \big(I - P(\ell) + \Pi(\ell)\big)^{-1}r(\ell)$; thus

$$\big(\hat{\pi}_n(\ell) - \pi(\ell)\big)r(\ell) = \hat{\pi}_n(\ell)\big(\hat{P}_n(\ell) - P(\ell)\big)\tilde{g}(\ell), \tag{59}$$

as desired.

Now for $\ell = 1, 2$, and $x \in S$, define

$$r(\ell, x, y) = \int_{\mathbb{R}} z F(dz, x, y, \ell). \tag{60}$$

417 Recall that $\hat{\alpha}_n = \hat{\pi}_n(2)\hat{r}_n(2) - \hat{\pi}_n(1)\hat{r}_n(1)$. We can write:

$$\hat{\pi}_n(\ell)\hat{r}_n(\ell) - \pi(\ell)r(\ell) \tag{61}$$

$$= \big(\hat{\pi}_n(\ell) - \pi(\ell)\big)r(\ell) + \hat{\pi}_n(\ell)\big(\hat{r}_n(\ell) - r(\ell)\big) \tag{62}$$

$$= \hat{\pi}_n(\ell)\big(\hat{P}_n(\ell) - P(\ell)\big)\tilde{g}(\ell) + \hat{\pi}_n(\ell)\big(\hat{r}_n(\ell) - r(\ell)\big) \tag{63}$$

$$= \sum_{x \in S} \hat{\pi}_n(\ell, x) \frac{\sum_{j=1}^n I(X_{j-1} = x, A_{j-1} = \ell)\big[\tilde{g}(\ell, X_j) - \big(P(\ell)\tilde{g}(\ell)\big)(X_{j-1})\big]}{\max\{\Gamma_n(\ell, x), 1\}} \tag{64}$$

$$+ \sum_{x \in S} \hat{\pi}_n(\ell, x) \frac{\sum_{j=0}^{n-1} I(X_j = x, A_j = \ell)\big(R_{j+1} - r(\ell, x)\big)}{\max\{\Gamma_n(\ell, x), 1\}}$$

$$= \sum_{x \in S} \hat{\pi}_n(\ell, x) \frac{\sum_{j=1}^n D_j(\ell, x)}{\max\{\Gamma_n(\ell, x), 1\}} \tag{65}$$

418 where

$$D_j(\ell, x) := I(X_{j-1} = x, A_{j-1} = \ell)\big[\tilde{g}(\ell, X_j) - \big(P(\ell)\tilde{g}(\ell)\big)(X_{j-1}) + R_j - r(\ell, X_{j-1})\big]. \tag{66}$$

419 Note that for each $\ell, x$, $D_j(\ell, x)$ is a martingale difference sequence adapted to $\mathcal{G}_j$.

420 For deterministic $w(\ell) = (w(\ell, x) : x \in S), \ell = 1, 2$, consider

$$T_n = \frac{1}{\sqrt{n}} \sum_{j=1}^n \sum_{\ell=1}^2 \sum_{x \in S} D_j(\ell, x)w(\ell, x) \tag{67}$$

$$\triangleq \frac{1}{\sqrt{n}} \sum_{j=1}^n D_j, \tag{68}$$

421 where

$$D_j = \sum_{\ell=1}^2 \sum_{x \in S} D_j(\ell, x)w(\ell, x). \tag{69}$$

422 The $D_j$'s are martingale differences adapted to $(\mathcal{G}_j : j \geq 0)$. Since they are bounded by
423 $2\max\{|\tilde{g}(\ell, x)| : x \in S, \ell = 1, 2\} < \infty$ (since $r(\ell, x)$ is finite), the following conditional Lin-
424 deberg's condition holds (Eq. (3.7) of [10]):

$$\text{for all } \epsilon > 0, \quad \sum_{j=1}^n \frac{1}{n} E\{D_j^2 I(|D_j| > \epsilon)|\mathcal{G}_{j-1}\} \xrightarrow{p} 0. \tag{70}$$

425 Furthermore,

$$\frac{1}{n} \sum_{j=1}^n E\{D_j^2|\mathcal{G}_{j-1}\} = \frac{1}{n} \sum_{\ell=1}^2 \sum_{x \in S} \sum_{j=0}^{n-1} I(X_j = x, A_j = \ell)\sigma^2(\ell, x)w^2(\ell, x) \tag{71}$$

$$= \sum_{\ell=1}^2 \sum_{x \in S} \sigma^2(\ell, x)w^2(\ell, x)\frac{\Gamma_n(\ell, x)}{n} \tag{72}$$

$$\xrightarrow{p} \sum_{\ell=1}^2 \sum_{x \in S} \sigma^2(\ell, x)w^2(\ell, x)\gamma(\ell, x) \triangleq \eta^2, \tag{73}$$

426 since $A$ is assumed to be a TAR policy. We therefore conclude that (by Corollary (3.1) of [10])

$$T_n \Rightarrow \sum_{\ell=1}^2 \sum_{x \in S} \sigma(\ell, x)w(\ell, x)\sqrt{\gamma(\ell, x)}G(\ell, x) \quad (stably) \tag{74}$$

427 as $n \to \infty$.[3]

428 Stable weak convergence implies that the following convergence of characteristic functions holds as
429 well:

$$E\bigg\{ \exp\Big(iT_n + i\sum_{\ell=1}^{2}\sum_{x\in S}\tilde{w}(\ell,x)\gamma(\ell,x)\Big)\bigg\} \tag{75}$$

$$\to E\bigg\{ \exp\Big(i\sum_{\ell=1}^{2}\sum_{x\in S}w(\ell,x)G(\ell,x)\sigma(\ell,x)\sqrt{\gamma(\ell,x)} + i\sum_{\ell=1}^{2}\sum_{x\in S}\tilde{w}(\ell,x)\gamma(\ell,x)\Big)\bigg\} \tag{76}$$

430 as $n \to \infty$, for any deterministic choice of $\tilde{w}(\ell) = (\tilde{w}(\ell,x) : x \in S), j = 1, 2$. The Cramer-Wold
431 device therefore implies that

$$\left(\frac{\sum_{j=1}^{n}D_j(\ell,x)}{\sqrt{n}}, \gamma(\ell,x) : x \in S, \ell = 1,2\right) \Rightarrow \left(\sigma(\ell,x)\sqrt{\gamma(\ell,x)}G(\ell,x), \gamma(\ell,x) : x \in S, \ell = 1,2\right) \tag{77}$$

432 as $n \to \infty$. Consequently, since the $\gamma(\ell,x)$'s are almost surely positive,

$$\left(\frac{\sum_{j=1}^{n}D_j(\ell,x)}{\sqrt{n}\gamma(\ell,x)} : x \in S, \ell = 1,2\right) \Rightarrow \left(\frac{\sigma(\ell,x)G(\ell,x)}{\sqrt{\gamma(\ell,x)}} : x \in S, \ell = 1,2\right) \tag{78}$$

433 as $n \to \infty$. Because $\frac{\Gamma_n(\ell,x)}{n\gamma(\ell,x)} \xrightarrow{P} 1$ as $n \to \infty$, Slutsky's lemma implies that

$$\sqrt{n}\left(\frac{\sum_{j=1}^{n}D_j(\ell,x)}{\Gamma_n(\ell,x)} : x \in S, \ell = 1,2\right) \tag{79}$$

$$\Rightarrow \left(\frac{\sigma(\ell,x)G(\ell,x)}{\sqrt{\gamma(\ell,x)}} : x \in S, \ell = 1,2\right) \tag{80}$$

434 as $n \to \infty$. Finally, Result 2, (80), and another application of Slutsky's lemma imply that

$$\sqrt{n}\left[\sum_{x\in S}\hat{\pi}_n(1,x)\frac{\sum_{j=1}^{n}D_j(1,x)}{\Gamma_n(1,x)} - \sum_{x\in S}\pi_n(2,x)\frac{\sum_{j=1}^{n}D_j(2,x)}{\Gamma_n(2,x)}\right] \tag{81}$$

$$\Rightarrow \sum_{x\in S}\frac{\pi(1,x)\sigma(1,x)G(1,x)}{\sqrt{\gamma(1,x)}} - \sum_{x\in S}\frac{\pi(2,x)\sigma(2,x)G(2,x)}{\sqrt{\gamma(2,x)}} \tag{82}$$

435 as $n \to \infty$, proving the Theorem. ∎

436 **Proof of Corollary 10.** Note that the Skorohod representation theorem together with Fatou's lemma
437 applied to (29) yields the following:

$$\liminf_{n\to\infty} n\,\mathrm{Var}(\hat{\alpha}_n - \alpha) \geq \sum_{\ell=1,2}\sum_{x\in S}\pi^2(\ell,x)\sigma^2(\ell,x)E\left\{\frac{1}{\gamma(\ell,x)}\right\}. \tag{83}$$

438 Using Jensen's inequality on the right hand side of (83), we obtain the result in (30), as required.
439 (Note that $E\{\gamma(\ell,x)\} > 0$ for all $\ell, x$ since we assumed the policy limits are almost surely positive.)
440 ∎

441 **Proof of Corollary 11.**

442 First we show the following limits hold:

$$\lim_{n\to\infty} E\left\{\sup_{\ell=1,2;x\in S}\left|\frac{\hat{\pi}_n(\ell,x)}{\max\{\Gamma_n(\ell,x),1\}/n} - \frac{\pi(\ell,x)}{\gamma(\ell,x)}\right|\right\} = 0; \tag{84}$$

$$\lim_{n\to\infty} E\left\{\sup_{\ell=1,2;x\in S}\left|\frac{\hat{\pi}_n(\ell,x)}{\max\{\Gamma_n(\ell,x),1\}/n} - \frac{\pi(\ell,x)}{\gamma(\ell,x)}\right|^2\right\} = 0. \tag{85}$$

We know from Proposition 6 that $\hat{\pi}_n \xrightarrow{p} \pi_n$ for all $\ell, x$. Further, we know from the definition of policy limits that $\Gamma(\ell, x)/n \xrightarrow{p} \gamma(\ell, x)$ for all $\ell, x$. Thus the vector $(\hat{\pi}_n, \Gamma_n/n)$ converges in probability to the vector $(\pi, \gamma)$. Use the Skorohod representation theorem to construct a joint probability space on which these limits hold almost surely. Then note that each of the terms inside the expectations are bounded in (84)-(85), so the desired results hold by the bounded convergence theorem.

For the next steps, we use the same definitions as in the proof of Theorem 9, and refer the reader there for the relevant notation. In particular, we define $D_j(\ell, x)$ as in that proof, and use the relationship in (65). We make the following two definitions:

$$Y_n(\ell) = \hat{\pi}_n(\ell)\hat{r}_n(\ell) - \pi(\ell)r(\ell) = \sum_{j=1}^{n} \sum_{x \in S} \frac{\hat{\pi}_n(\ell, x)}{\max\{\Gamma_n(\ell, x), 1\}/n} \cdot \frac{D_j(\ell, x)}{n};$$

$$Z_n(\ell) = \sum_{j=1}^{n} \sum_{x \in S} \frac{\pi(\ell, x)}{\gamma(\ell, x)} \cdot \frac{D_j(\ell, x)}{n}.$$

Note that $\hat{\alpha}_n - \alpha = Y_n(2) - Y_n(1)$. The main remaining step in our proof is to show that we can compute the scaled asymptotic variance of $Z_n(2) - Z_n(1)$, and to use this to upper bound the scaled asymptotic variance of $Y_n(2) - Y_n(1)$.

We now show the following limit holds:

$$\lim_{n \to \infty} \text{Var}(\sqrt{n}(Z_n(2) - Z_n(1))) = \sum_{x \in S} \frac{\pi^2(\ell, x)\sigma^2(\ell, x)}{\gamma(\ell, x)}. \tag{86}$$

Observe that $Z_n(\ell)$ is a weighted sum of martingale differences; thus we use orthogonality of martingale differences again. In particular, $E\{Z_n(\ell)\} = 0$ for all $n$. Thus $\text{Var}(\sqrt{n}(Z_n(2) - Z_n(1))) = nE\{(Z_n(2) - Z_n(1))^2\}$. Observe that:

$$Z_n(1)Z_n(2) = \sum_{j=1}^{n} \sum_{k=1}^{n} \sum_{x,y \in S} \frac{\pi(1, x)\pi(2, y)}{\gamma(1, x)\gamma(2, y)} \frac{D_j(1, x)D_k(2, y)}{n}.$$

We show that $E\{D_j(1, x)D_k(2, y)\} = 0$. If $j = k$, then the product $D_j(1, x)D_j(2, y) = 0$ since only one of the two chains $\ell = 1, 2$ can be run at time $k$. If $j > k$, then the tower property of conditional expectations is applied as usual to give:

$$E\{E\{D_j(1, x)|\mathcal{G}_k\}D_k(2, x)\} = 0.$$

The same holds of course if $j < k$. Thus we have $E\{Z_n(1)Z_n(2)\} = 0$ for all $n$. Finally, using (71) with $w(1, x) = \pi(1, x)/\gamma(1, x)$ and $w(2, x) = 0$, together with the Skorohod representation theorem and the bounded convergence theorem, it follows that:

$$E\{nZ_n(1)^2\} \to \sum_{x \in S} \frac{\pi^2(1, x)\sigma^2(1, x)}{\gamma(1, x)}.$$

(Use of bounded convergence here requires assuming boundedness of rewards.) An analogous result holds for the limit of $E\{nZ_n(2)^2\}$. Combining these steps, we obtain (86).

Finally, we can establish the following upper bound:

$$\limsup_{n \to \infty} n \text{Var}(\hat{\alpha}_n - \alpha_n) \le \sum_{\ell=1,2} \sum_{x \in S} \frac{\pi^2(\ell, x)\sigma^2(\ell, x)}{\gamma(\ell, x)}. \tag{87}$$

To prove this we upper bound the variance of $Y_n(2) - Y_n(1)$ in terms of the variance of $Z_n(2) - Z_n(1)$. Note that $\text{Var}(Y_n(2) - Y_n(1)) \le E\{(Y_n(2) - Y_n(1))^2\}$. Further, because $D_j(\ell, x)$ are bounded, there exist constants $M_1, M_2$ such that:

$$(Y_n(2) - Y_n(1))^2 \le (Z_n(2) - Z_n(1))^2 + M_1 \sup_{\ell=1,2; x \in S} \left| \frac{\hat{\pi}_n(\ell, x)}{\max\{\Gamma_n(\ell, x), 1\}/n} - \frac{\pi(\ell, x)}{\gamma(\ell, x)} \right|$$

$$+ M_2 \sup_{\ell=1,2; x \in S} \left| \frac{\hat{\pi}_n(\ell, x)}{\max\{\Gamma_n(\ell, x), 1\}/n} - \frac{\pi(\ell, x)}{\gamma(\ell, x)} \right|^2.$$

470    Taking expectations on both sides, and applying Steps 2 and 3, establishes (87). Combining (87) with
471    (30) yields the desired result (note that $E\{\gamma(\ell, x)\} = \gamma(\ell, x)$ since the policy limits are almost surely
472    constant). ∎

473    **Proof of Theorem 13.** First, we show that (33)-(34) has a unique optimal solution $\kappa^*$, with entries
474    that are all positive. It is straightforward to see that the solution to this problem will be positive in all
475    coordinates, since the objective function approaches infinity as any $\kappa(\ell, x)$ approaches zero (as all
476    $\sigma(\ell, x)$ are positive). Further, note that the objective function is strictly convex and $\mathcal{K}$ is convex and
477    compact, and thus there must be a unique solution $\kappa^* \in \mathcal{K}$ to the optimization problem (33)-(34).

478    Next, we show that the limit inferior of the scaled asymptotic variance of the MLE under any TAR
479    policy with positive policy limits is bounded below by the optimal value of (33)-(34). This follows
480    by applying Corollary 10. In particular, from Remark 5, we know $\gamma$ is a probability measure over the
481    set $\mathcal{K}$ (cf. Definition 3). The set $\mathcal{K}$ is convex and compact, and so $\kappa = E\{\gamma\} \in \mathcal{K}$. In particular, as a
482    consequence by applying (30) we conclude that the optimal value of (33)-(34) is a lower bound to
483    $\liminf_{n \to \infty} \text{Var}(\hat{\alpha}_n - \alpha)$.

484    Finally, the fact that (35) holds follows from Corollary 11. The stationary Markov policy $A^*$ defined
485    via (20) has the constant policy limit $\kappa^*$ (cf. Remark 5), so it is efficient. The theorem follows. ∎

# D    Pseudocode for OnlineETI

487    The pseudocode for OnlineETI is prsented as Algorithm 1.

# E    Proofs: Section 6

489    **Proof of Theorem 14.** We establish that for OnlineETI there holds:

$$\frac{1}{n}\Gamma_n(\ell, x) \xrightarrow{p} \kappa^*(\ell, x), \tag{88}$$

490    where $\kappa^*$ is the solution to (33)-(34).

491    First, note that the forced exploration (i.e., the $M_n(x)^{-1/2}$ term in the definition of $\hat{p}_n(\ell, x)$) ensures
492    that $\Gamma_n(\ell, x) \to \infty$ almost surely for all $\ell, x$. To see this, note first that as long as $M_n(x) \to \infty$
493    almost surely, it must be the case that $\Gamma_n(\ell, x) \to \infty$ for $\ell = 1, 2$ almost surely as well, due to the
494    forced exploration term, the fact that $\sum_{k \geq 1} k^{-1/2}$ diverges, and the Borel-Cantelli Lemma. Since
495    the state space is finite, almost surely, there exists at least one state $x'$ that is visited infinitely often.
496    Thus almost surely, all states reachable from $x'$ in one step under either $P(1)$ or $P(2)$ must be visited
497    infinitely often as well. The same argument applies to those states, and so on. Since the state space is
498    finite, and both $P(1)$ and $P(2)$ are irreducible, this process exhausts all the states, and we conclude
499    $M_n(x) \to \infty$ almost surely for all $x \in S$.

500    Next we show that for all $\ell, x, y$, $\hat{P}_n(\ell, x, y)$ converges to $P(\ell, x, y)$ almost surely. For each
501    $\ell, x$, it is convenient to define $T_m(\ell, x) = \inf\{n : \Gamma_n(\ell, x) = m\}$. By the standard strong law
502    of large numbers, it follows that $\hat{P}_{T_m(\ell, x)}(\ell, x, y) \to P(\ell, x, y)$ almost surely; this is because
503    $\hat{P}_{T_m(\ell, x)}(\ell, x, y)$ is the sample average of $m$ independent Bernoulli random variables, each with
504    success probability $P(\ell, x, y)$. Now observe that for $n$ such that $T_m(\ell, x) \leq n < T_{m+1}(\ell, x)$,
505    $\hat{P}_n(\ell, x, y) = \hat{P}_{T_m(\ell, x)}(\ell, x, y)$; i.e., between successive visits to state $x$ in which policy $\ell$ is sampled,
506    $\hat{P}_n(\ell, x, y)$ remains constant. It follows therefore that $\hat{P}_n(\ell, x, y) \to P(\ell, x, y)$ almost surely as well.

507    We now use a compactness argument analogous to that used to establish (23) to show that $\hat{\pi}_n(\ell) \to$
508    $\pi(\ell)$ almost surely. Let $J$ be the first $n$ at which $\hat{P}_n(\ell)$ is irreducible for both $\ell = 1, 2$. The time
509    $J$ is almost surely finite, because both chains are sampled with equal probability until time $J$, and
510    because $P(\ell)$ is irreducible for $\ell = 1, 2$. Thus for the remainder of our argument, we condition on
511    the almost sure event $J < \infty$. Next, consider any subsequence $\{n_k\}$ along which, almost surely,
512    $\hat{\pi}_{n_k}(\ell) \to \pi'(\ell)$. (Note that in general, this is a random subsequence.) Since $\hat{\pi}_{n_k}(\ell)\hat{P}_{n_k}(\ell) = \hat{\pi}_{n_k}(\ell)$
513    for all $k$, almost sure convergence of $\hat{P}_n(\ell)$ implies that $\pi'(\ell)P(\ell) = \pi'(\ell)$. Thus $\pi'(\ell) = \pi(\ell)$
514    almost surely. Since this is almost surely true for every convergent subsequence, we conclude that
515    $\hat{\pi}_n(\ell) \to \pi(\ell)$ almost surely, as required.

---

**Algorithm 1** OnlineETI (Online Experimentation with Temporal Interference)

---

1: **procedure** EXPERIMENT(initial state $x_0$)
2:     Set initial state $X_0 = x_0$
3:     Initialization: For $\ell = 1, 2$, $x, y \in S$, set $\hat{P}_0(\ell, x, y) = \frac{1}{|S|}$; $\Gamma_0(\ell, x) = 0$; $\Phi_0(\ell, x, y) = 0$;
4:         $\Theta_0(\ell, x) = 0$; $\Psi_0(\ell, x) = 0$; $\Upsilon_0(\ell, x, y) = 0$; $\hat{r}_0(\ell, x) = 0$; $\hat{s}_0(\ell, x, y) = 0$;
5:         $\hat{t}_0(\ell, x, y) = 0$; $\hat{\pi}_0(\ell, x) = 0$; $\hat{p}_0(\ell, x) = 0.5$
6:     **for** $n = 1, 2, \ldots$ **do**
7:         Set $A_{n-1} = \ell$ with probability $\hat{p}_{n-1}(\ell, x)$, i.e.:
8:             $A_{n-1} = 1$ if $U_{n-1} \leq \hat{p}_{n-1}(1, x)$, and $A_{n-1} = 2$ otherwise
9:         Run chain $A_{n-1}$, and obtain reward $R_n$ and new state $X_n$
10:        For all $\ell = 1, 2$, $x, y \in S$:
11:           $\Gamma_n(\ell, x) \leftarrow \Gamma_{n-1}(\ell, x) + I(X_{n-1} = x, A_{n-1} = \ell)$
12:           $\Phi_n(\ell, x, y) \leftarrow \Phi_{n-1}(\ell, x, y) + I(X_{n-1} = x, X_n = y, A_{n-1} = \ell)$
13:           $\Theta_n(\ell, x) \leftarrow \Theta_{n-1}(\ell, x) + I(X_{n-1} = x, A_{n-1} = \ell)R_n$
14:           $\Psi_n(\ell, x, y) = \Psi_{n-1}(\ell, x, y) + I(X_{n-1} = x, X_n = y, A_{n-1} = \ell)R_n$
15:           $\Upsilon_n(\ell, x, y) \leftarrow \Upsilon_{n-1}(\ell, x, y) + I(X_{n-1} = x, X_n = y, A_{n-1} = \ell)R_n^2$
16:           $\hat{P}_n(\ell, x, y) \leftarrow \frac{\Phi_n(\ell, x, y)}{\max\{\Gamma_n(\ell, x), 1\}}$
17:        **if** for both $\ell = 1, 2$, $\hat{P}_n(\ell)$ is irreducible **then**
18:           Set $\hat{\pi}_n(\ell)$ to be the unique steady state distribution of $\hat{P}_n(\ell)$
19:           For $\ell = 1, 2$ and $x, y \in S$:
20:              $\hat{\Pi}_n(\ell) \leftarrow e\hat{\pi}_n(\ell)$
21:              $\hat{\tilde{g}}_n(\ell, x) \leftarrow \left(I - \hat{P}_n(\ell) + \hat{\Pi}_n(\ell)\right)^{-1}\hat{r}_n(\ell)$
22:              $\hat{r}_n(\ell, x) \leftarrow \frac{\sum_{y \in S}\Psi_n(\ell, x, y)}{\max\{\Gamma_n(\ell, x), 1\}}$
23:              $\hat{s}_n(\ell, x, y) \leftarrow \frac{\Psi_n(\ell, x, y)}{\max\{\Phi_n(\ell, x, y), 1\}}$
24:              $\hat{t}_n(\ell, x, y) \leftarrow \frac{\Upsilon_n(\ell, x, y)}{\max\{\Phi_n(\ell, x, y), 1\}}$
25:              $\hat{\sigma}_n^2(\ell, x) \leftarrow \sum_{y \in S}\hat{P}_n(\ell, x, y)[\hat{\tilde{g}}_n(\ell, y) - \sum_{z \in S}\hat{P}_n(\ell, x, z)\hat{\tilde{g}}_n(\ell, z)]^2$
26:              $+ \sum_{y \in S}\hat{P}_n(\ell, x, y)\left(\hat{t}_n(\ell, x, y) - \hat{s}_n(\ell, x, y)^2\right)$
27:           Choose any $\hat{\kappa}_n$ in $\arg\inf_{\hat{\kappa} \in \mathcal{K}} \sum_{\ell=1}^{2}\sum_{x \in S}\frac{\hat{\pi}_n^2(\ell, x)\hat{\sigma}_n^2(\ell, x)}{\hat{\kappa}_n(\ell, x)}$
28:           For all $x \in S$, $M_n(x) \leftarrow \Gamma_n(1, x) + \Gamma_n(2, x)$
29:           **if** $\hat{\kappa}_n(1, x) + \hat{\kappa}_n(2, x) > 0$ and $M_n(x) > 0$ **then**
30:              $\hat{p}_n(\ell, x) \leftarrow (1 - M_n(x)^{-1/2})\left(\frac{\hat{\kappa}_n(\ell, x)}{\hat{\kappa}_n(1, x) + \hat{\kappa}_n(2, x)}\right)$
31:              $+ \frac{1}{2}M_n(x)^{-1/2}$ for $\ell = 1, 2$, $x \in S$
32:           **else**
33:              $\hat{p}_n(\ell, x) = 0.5$ for $\ell = 1, 2$, $x \in S$
34:           $\hat{\alpha}_n \leftarrow \hat{\pi}_n(2)\hat{r}_n(2) - \hat{\pi}_n(1)\hat{r}_n(1)$
35:        **else**
36:           $\hat{p}_n(\ell, x) \leftarrow 0.5$
37:           $\hat{\alpha}_n \leftarrow 0$

---

Because rewards are bounded, and thus in particular have finite moments, an argument analogous to that above for $\hat{P}_n$ establishes that almost surely we have:

$$\hat{r}_n(\ell, x) \to r(\ell, x)$$

and

$$\hat{t}_n(\ell, x, y) - \hat{s}_n^2(\ell, x, y)^2 \to \mathrm{Var}(R_1 | A_0 = \ell, X_0 = x, X_1 = y).$$

When $J < \infty$, since each $\hat{P}_n(\ell)$ is irreducible, it follows that $\left(I - \hat{P}_n(\ell) + \hat{\Pi}_n(\ell)\right)^{-1}$ exists. By continuity, conditioning on $J < \infty$, we have:

$$\hat{\tilde{g}}(\ell, x) \to \tilde{g}(\ell, x)$$

almost surely as well, and thus:

$$\hat{\sigma}^2(\ell, x) \to \sigma^2(\ell, x)$$

522    almost surely.

523    We now establish almost sure convergence of $\hat{\kappa}_n$ to $\kappa^*$. To do this, for a distribution $\tilde{\pi}$ on the state
524    space $S$ and a nonnegative vector $\tilde{\sigma}$, define the correspondence $K(\tilde{\pi}, \tilde{\sigma})$ to be the set of minimizers of
525    $\sum_{\ell=1,2} \sum_{x \in S} \tilde{\pi}^2(\ell, x) \tilde{\sigma}^2(\ell, x) / \kappa(\ell, x)$ over $\kappa \in \mathcal{K}$; recall that $\mathcal{K}$ is compact so this correspondence
526    is nonempty everywhere. Further, observe that if $\tilde{\pi}$ and $\tilde{\sigma}$ are positive in all coordinates, then the
527    minimizer is unique, i.e., $K$ is a function. Then by Lemma 15 below, $K$ is continuous in $\tilde{\pi}$ and $\tilde{\sigma}$
528    when they are both positive in all coordinates. Since $\hat{\pi}_n(\ell) \to \pi(\ell)$ and $\hat{\sigma}_n^2(\ell, x) \to \sigma^2(\ell, x)$ almost
529    surely, and both limits are positive in all coordinates, it follows that $K(\hat{\pi}_n, \hat{\sigma}_n) \to K(\pi, \sigma) = \kappa^*$
530    almost surely, and thus $\hat{\kappa}_n \to \kappa^*$ almost surely.

531    In particular, we thus know that almost surely, $\hat{\kappa}_n(\ell, x) > 0$ for all sufficiently large $n$. As a result, it
532    follows that $\hat{p}_n(\ell, x) \to p^*(\ell, x)$ almost surely, where:

$$p^*(\ell, x) = \frac{\kappa^*(\ell, x)}{\kappa^*(1, x) + \kappa^*(2, x)}.$$

533    To complete the proof, we require some additional notation. We define the following stochastic
534    matrix:

$$Q(x, y) = p^*(1, x) P(1, x, y) + p^*(2, x) P(2, x, y).$$

535    Note that this matrix is irreducible, and because $\kappa^* \in \mathcal{K}$, we can easily see that $Q$ has the unique
536    stationary distribution given by:

$$\zeta^*(x) = \kappa^*(1, x) + \kappa^*(2, x).$$

537    (See also the discussion in Remark 5.)

538    In addition, we define:

$$\hat{Q}_n(x, y) = \frac{\sum_{j=1}^{n} I(X_{j-1} = x, X_j = y)}{\max\{M_n(x), 1\}}.$$

539    Observe that $\hat{Q}_n$ is a stochastic matrix.

540    We now show that $\hat{Q}_n \xrightarrow{p} Q$. We rewrite $\hat{Q}_n(x, y)$ as follows:

$$\hat{Q}_n(x, y) = \sum_{\ell=1,2} \hat{P}_n(\ell, x, y) \cdot \frac{\sum_{j=1}^{n} I(X_{j-1} = x, A_{j-1} = \ell)}{\max\{M_n(x), 1\}}. \tag{89}$$

541    For each $x$ and $m$, let $S_m(x) = \inf\{n \geq 0 : M_n(x) = m\}$; this is the time step at which the $m$'th
542    visit to $x$ takes place. Further, define $\tilde{A}_m = A_{S_m(x)}$; this is the policy sampled at the $m$'th visit to $x$.
543    Let $\mathcal{H}_m(x) = \sigma((X_j, U_j, V_j, j < S_m(x); X_{S_m(x)}))$ be the sigma field generated by randomness up
544    to the $m$'th visit to $x$, but prior to the policy being chosen. Finally, let $\hat{q}_m(\ell, x) = \hat{p}_{S_m(x)}(\ell, x)$. Now
545    observe that when $M_n(x) = m \geq 1$, we have:

$$\frac{\sum_{j=1}^{n} I(X_{j-1} = x, A_{j-1} = \ell)}{\max\{M_n(x), 1\}} = \frac{\sum_{i=1}^{m} I(\tilde{A}_i = \ell)}{m}$$
$$= \frac{\sum_{i=1}^{m} I(\tilde{A}_i = \ell) - \hat{q}_i(\ell, x)}{m} + \frac{\sum_{i=1}^{m} \hat{q}_i(\ell, x)}{m}.$$

546    The terms in the first sum on the right hand side of the previous expression form a martingale
547    difference sequence adapted to $\mathcal{H}_i$. Thus using orthogonality of martingale differences, we have:

$$\frac{1}{m^2} E\left\{ \left( \sum_{i=1}^{m} I(\tilde{A}_i = \ell) - \hat{q}_i(\ell, x) \right)^2 \right\} \leq \frac{1}{4m},$$

548    which approaches zero as $m \to \infty$. By Chebyshev's inequality, it follows that:

$$\frac{\sum_{i=1}^{m} I(\tilde{A}_i = \ell) - \hat{q}_i(\ell, x)}{m} \xrightarrow{p} 0$$

as $m \to \infty$. On the other hand, note that since $M_n(x) \to \infty$ almost surely, we also know that $S_m(x) \to \infty$ as $m \to \infty$ almost surely. Thus it follows that:

$$\frac{\sum_{i=1}^m \hat{q}_i(\ell, x)}{m} \to p^*(\ell, x)$$

almost surely as $m \to \infty$, and thus in probability as well. Combining these insights, we conclude that:

$$\frac{\sum_{j=1}^n I(X_{j-1} = x, A_{j-1} = \ell)}{\max\{M_n(x), 1\}} \xrightarrow{p} p^*(\ell, x)$$

as $n \to \infty$, and so returning to (89), we find that:

$$\hat{Q}_n(x, y) \xrightarrow{p} \sum_{\ell=1,2} p^*(\ell, x) P(\ell, x, y) = Q(x, y).$$

Next, observe that:

$$\frac{M_n(x)}{n} = \frac{\sum_{j=1}^n I(X_j = x)}{n} + \frac{I(X_0 = x) - I(X_n = x)}{n}$$

$$= \left( \sum_{y \in S} \hat{Q}_n(x, y) \cdot \frac{\max\{M_n(y), 1\}}{n} \right) + O_p\left(\frac{1}{n}\right).$$

Since $M_n(x) \to \infty$ almost surely, in what follows we condition on $M_n(x) \geq 1$ for all $x$ and thus ignore the "max" on the right hand side in the preceding expression. Note that for all $n$, $\sum_{x \in S} M_n(x) = n$. Thus using a compactness argument analogous to that used to establish (23), it follows that:

$$\frac{M_n(n)}{n} \xrightarrow{p} \zeta^*(x).$$

We can now complete the proof of the theorem. We have:

$$\frac{1}{n}\Gamma_n(\ell, x) = \frac{1}{n} \sum_{j=0}^{n-1} I(X_j = x, A_j = \ell)$$

$$= \frac{1}{n} \sum_{j=0}^{n-1} I(X_j = x) p^*(\ell, x) + \frac{1}{n} \sum_{j=0}^{n-1} I(X_j = x)\big(\hat{p}_j(\ell, x) - p^*(\ell, x)\big)$$

$$+ \frac{1}{n} \sum_{j=0}^{n-1} I(X_j = x)\big(I(A_j = \ell) - \hat{p}_j(\ell, x)\big) \tag{90}$$

Because $I(X_j = x)\big(I(A_j = \ell) - \hat{p}_j(\ell, x)\big)$ is a martingale difference measurable with respect to $\mathcal{G}_j$, orthogonality of martingale differences implies that

$$E\left\{ \left( \frac{1}{n} \sum_{j=1}^n I(X_j = x)\big(I(A_j = \ell) - \hat{p}_j(\ell, x)\big) \right)^2 \right\} \tag{91}$$

$$\leq E\left\{ \frac{1}{4} \cdot \frac{1}{n^2} \Gamma_n(\ell, x) \right\} \leq \frac{1}{4n} \tag{92}$$

$$\to 0 \tag{93}$$

as $n \to \infty$. Therefore, by Chebyshev's inequality

$$\frac{1}{n} \sum_{j=0}^{n-1} I(X_j = x)\big(I(A_j = \ell) - \hat{p}_j(\ell, x)\big) \xrightarrow{p} 0 \tag{94}$$

as $n \to \infty$. Also, since $\hat{p}_n(\ell, x) \to p^*(\ell, x)$ almost surely, we have:

$$\frac{1}{n} \sum_{j=0}^{n-1} I(X_j = x)\big(\hat{p}_j(\ell, x) - p^*(\ell, x)\big) \xrightarrow{p} 0. \tag{95}$$

564 Finally,

$$\frac{1}{n} \sum_{j=0}^{n-1} I(X_j = x) p^*(\ell, x) = \frac{p^*(\ell, x) M_n(\ell, x)}{n} \xrightarrow{p} p^*(\ell, x) \zeta^*(\ell, x).$$

565 Combining the preceding results, we conclude that as $n \to \infty$ in (90), we have

$$\frac{1}{n} \Gamma_n(\ell, x) \xrightarrow{p} \zeta^*(\ell, x) p^*(\ell, x) = \kappa^*(\ell, x) \tag{96}$$

566 as $n \to \infty$, completing the proof of the theorem. ∎

567 **Lemma 15** *Suppose that the set $X$ is compact, the set $\Theta$ is open, and the real-valued function $f(\theta, x)$*
568 *is continuous on the domain $\Theta \times X$. Suppose further that for every $\theta \in \Theta$, there exists a unique*
569 *$x^*(\theta) = \arg\min_{x \in X} f(\theta, x)$. Then $x^*(\theta)$ is continuous in $\theta$.*

570 **Proof.** Suppose that $\theta^{(n)} \to \theta$. For all $n$ we have:

$$f(\theta^{(n)}, x^*(\theta^{(n)})) \leq f(\theta^{(n)}, x^*(\theta)). \tag{97}$$

571 Since $X$ is compact, let $\{n_k\}$ be a subsequence such that $x^*(\theta^{(n_k)}) \to x'$ as $k \to \infty$. Taking limits
572 on both sides of (97) along the sequence $\{n_k\}$, we obtain:

$$f(\theta, x') \leq f(\theta, x^*(\theta)).$$

573 Since $x^*(\theta)$ is unique, this is only possible if $x' = x^*(\theta)$. Since every convergent subsequence must
574 have the limit $x'$, we conclude that $x^*(\theta^{(n)}) \to x^*(\theta)$ as $n \to \infty$, as required. ∎

## Footnotes

[3] If a sequence of random variables $Y_n$ on a probability space $(\Omega, \mathcal{F}, P)$ converges weakly to $Y$, we say the convergence is *stable* if for all continuity points $y$ of the cumulative distribution function of $Y$ and for all measurable events $E$, the limit $\lim_{n\to\infty} P(\{Y_n \leq y\} \cap E) = Q_y(E)$ exists, and if $Q_y(E) \to P(E)$ as $y \to \infty$.