[Reviews · NeurIPS 2020]

Review 1

Summary and Contributions: The authors study the problem of comparing a treatment and control policy under temporal interference. That is, when the initial condition in each interval of each policy is determined by the previous interval of the other policy. The goal is to find an optimal experimental design for this setting. Each policy is viewed as a Markov chain, and the experimental design problem is to estimate the difference in the steady state reward under the treatment and control Markov chains. Estimating unknowns are done using a nonparametric maximum likelihood estimator, which is shown to be consistent under a regularity requirement called time-average regularity (TAR). The main result of the paper is characterizing efficient TAR experiment design, i.e., those for which the MLE achieves asymptotically minimum variance among all TAR experiment designs. This characterization is used to construct an adaptive online experiment design for which the MLE achieves efficient and consistent estimations.

Strengths: - The paper is well written and the claims seem sound. - The work provides a condition and an approach for obtaining efficient, consistent experiment design for settings with temporal interference.

Weaknesses: - Can we interpret the results as follows: If the TAR assumption is satisfied with positive limits, and we use MLE, then temporal interference does not cause bias. If this interpretation is correct, then it would be illuminating if the authors provide the intuitive connection between the TAR assumption and temporal interference. - One of the important concerns regarding the work is the requirement of using MLE. It is not clear if the estimations that the authors have required are feasible if the state space is large. The next natural question is how robust the results are if we use other methods for estimation. This could have been shown by providing some simulations, which is a part missing from the manuscript. - Having some synthetic evaluations would have also been helpful in providing some intuition about the required sample size for obtaining desired performance and assessing the approach in finite horizon. - Is there any straight forward way to extent the results to the case of multiple treatments?

Correctness: Yes.

Clarity: Yes. A small comment: The term "policy" is used both for the Markov chains and the experiment design strategy. It can help the clarity if different terms are used.

Relation to Prior Work: Yes.

Reproducibility: Yes

Additional Feedback: After rebuttal: I thank the authors for their responses. My score remains unchanged.


Review 2

Summary and Contributions: This is a paper on experimental design with the following scenario: two policies are being run and compared by running them alternately for just a single run. There is usually temporal interference between the two policies being compared since the system is left in the state by the other policy when we switch. The authors analyze this situation as two Markov chains with a meta-policy describing how one switches between the two. They describe how to design an MLE for the two Markov chains. They study the asymptotic properties and prove a fundamental theorem (Thm. 9) which is a kind of central limit theorem which gives the weak convergence properties of the estimators. They use this result to define a convex optimization problem that characterises optimal experimental designs. They also give an online policy that allows them to construct the optimal design.

Strengths: This is in my opinion a striking and interesting result. Theorem 9 and Theorems 13 and 14 are very powerful and unexpected results. The mathematics is well described and quite sophisticated. The results seem very significant to me.

Weaknesses: The level of sophistication is quite high for a typical NeurIPS paper. I imagine that this is a difficult paper for most machine learning practitioners. I think there is a typo on line 117, should it be action "l" rather than "i"?

Correctness: Yes, the claims seem to be correct as far as I could tell.

Clarity: Yes.

Relation to Prior Work: Yes, previous work is discussed.

Reproducibility: Yes

Additional Feedback:


Review 3

Summary and Contributions: The paper considered an optimal experimental design for settings with temporal interference. In particular, the authors used a single run to estimate the performance of the systems under treatment and control policies. They tried to estimate the difference in the steady-state reward for the treatment and control Markov chains. They proposed asymptotically efficient policy using a maximum likelihood estimator.

Strengths: - The authors considered time-average regularity (TAR) policies and showed that MLE is consistent. - They proposed consistent adaptive experimental design where the estimations are obtained from MLE.

Weaknesses: ===After rebuttal=== I read the reviews and the rebuttal. It seems that the proposed algorithm might have complexities issues in large state space. Moreover, the current verion of the paper does not provide experimental results for the proposed method. Thus, I decided to keep my score unchanged. =============== - It is not clear what is the time complexity of optimization problem in Theorem 13. Since the proposed algorithm in Section 6 uses the minimizer of this optimization problem in each step, it is necessary to analyze the time complexity of this algorithm. Moreover, what is the size of the set K in Theorem 13? - Is it possible to charachterize what type of policies are time-average regular? - The proposed algorithm should be evaluted at least with synthetic data.

Correctness: It seems that the claims and the proposed algorithm are correct. However, I did not go in details of the proofs.

Clarity: The paper is generally well-written but it has some typos.

Relation to Prior Work: The authors mentioned the related work in Introduction section.

Reproducibility: Yes

Additional Feedback:


Review 4

Summary and Contributions: This paper considers design of an online policy selecting at each time step between two available procedures with the objective of estimating with minimal variance the difference between the long-term rewards obtained by the two distinct procedures. It provides a clean characterization of Maximum Likelihood Estimation for this long-term reward difference, introduces a class of policies (TAR) for which it characterizes the asymptotic variance of MLE estimation. It finally introduces onlineETI, an online policy achieving the optimal asymptotic variance in this estimation among all TAR policies.

Strengths: The problem addressed in this paper is highly relevant, and the approach proposed leads to clean insights. The theory is very neat, and the writing flows quite well.

Weaknesses: Since the paper is of theoretical nature, it would be strengthened by (short) discussion of how its insights can fuel practical experimental design.

Correctness: The claims appear to be correct.

Clarity: The writing is very clear, pleasant and easy to follow.

Relation to Prior Work: Yes to the best of my knowledge.

Reproducibility: Yes

Additional Feedback: References [5] and [14] appear incomplete. It would be nice to add, if possible, a discussion on optimality of estimation variance, not just among TAR +MLE policies, but among a broader set of sampling / estimation procedures. Indeed it seems plausible that the proposed approach is optimal more generally than currently stated. Another intriguing question is about possibility to achieve an even better reward than that of the two policies by allowing one to mix between the two policies at all times. A remark on this would be interesting to add.

[Author Response · NeurIPS 2020]

**Adaptive Experimental Design with Temporal Interference: A Maximum Likelihood Approach**

*Response to reviewers*

We are grateful to the referees for their thoughtful comments regarding our paper. Regarding typos and other detailed suggestions, we plan to incorporate those prior to submission of our camera ready version. Below we have also provided feedback in response to major comments and requests from the referees.

**Practical considerations**. Several referees commented on practical considerations (Reviewers 1, 3, and 4): in particular, the performance of maximum likelihood estimation (MLE) with large state spaces and on finite horizons, as well as its computational complexity. More broadly, we note that in followup work, we have developed an alternative approach to experimental design for temporal interference using sample average estimation (SAE). In this work, we study sampling strategies based on the *regenerative method*; together, SAE and regenerative policies ensure consistency while being practically implementable. The idea is that we commit to a single state (the *regeneration state*), and only allow ourselves to switch chains in this state. This method can be used together with sample average estimation (SAE) of rewards, rather than (MLE). Further, it requires no advance knowledge of the state space, other than the regeneration state. This is a much more practical, scalable solution; although it is not as sample efficient as the policy and MLE in our present paper, we can use similar techniques to obtain the optimal regenerative policy with SAE. Though we could not include this work due to space constraints, we will add some discussion of this extension to the paper to address concerns regarding practical implementation.

**Reviewer 1**. Regarding how to interpret our results, note that consistency follows under very general conditions: *any* sampling procedure that samples both experiments infinitely often in each state will ultimately yield consistent estimates via the MLE. By using the MLE instead of just averaging rewards obtained in runs of each chain, we avoid temporal interference completely. The main contribution of our paper is to provide a strong characterization of the *sample efficient* experimental design for the MLE.

Regarding practical considerations and state space complexity, note that our theory shows that for any finite state space our policy eventually outperforms any other TAR policy. Nevertheless, you are right that for a given sample size, our policy may be computationally complex; see our comments above on practical considerations.

Regarding multiple treatments, we conjecture that the optimal policy in the multiple treatments setting eventually looks essentially like ours, once the best and second best treatments have been identified.

**Reviewer 2**. Thank you for your review. Regarding making the paper more accessible to the Neurips community: we will plan to provide some more intuition for the main results in the final version.

**Reviewer 3**. Regarding the convex optimization problem in Theorem 13 and Section 6, we expect that in many applications the switching time is slow enough relative to computation time, so that standard convex optimization techniques can be employed. That said, we agree that computational complexity is an important practical issue; see our discussion above.

The set $\mathcal{K}$ is a closed, compact, convex polytope in $2|S|$-dimensional space (where $S$ is the state space).

Regarding what policies are time-average regular (TAR), we emphasize that TAR is a weak regularity requirement: it says that the fraction of time steps in which chain $\ell$ is sampled in state $x$ converges to a well-defined random variable (possibly deterministic). Virtually any reasonable policy will satisfy this requirement. Policies which, e.g., switch chains on exponentially increasing timescales will not be TAR, but such policies are not likely to be used in practice.

**Reviewer 4**. Regarding more general optimality results, we agree with your conjecture; this remains an important open direction. Note that for any fixed TAR policy, the MLE is asymptotically efficient given the samples collected by that policy. Further, our results show how to compute the optimal TAR policy when estimating via the MLE.

Regarding obtaining higher reward by combining the two chains, our work is entirely focused only on *estimation* of the difference in steady state rewards of the two chains, rather than *optimization* of the cumulative reward obtained. An interesting question concerns whether regret optimal policies can be designed to maximize the cumulative reward using only the two chains; this is an interesting reinforcement learning problem for future study.

**Other comments: Synthetic evaluation.** We agree with Reviewers 1 and 3 that synthetic evaluation would be valuable to add to our paper. Synthetic evaluation would primarily be valuable to study finite horizon performance, as our paper provides a full characterization of optimal asymptotic sample efficiency. We also emphasize that our paper is primarily a theoretical study of optimal experimental design in this setting; practical considerations can lead to different preferred designs (cf. our discussion above). Intuitively we believe the regenerative method leads to designs with improved finite horizon performance; we plan to carry out a synthetic study to compare and contrast finite horizon performance of various designs as part of our future work.

[Meta-Review · NeurIPS 2020]

The paper studied the online experimental design problem where there are temporal dependencies between the two control policies/treatments. The novelty of the problem setup and the theoretical analysis in the paper are appreciated by all the reviewers. Although the analysis is the main contribution, the paper would be much stronger if there are meaningful experiments on toy problems to showcase the performance the online MLE-based approach vs the standard experimental design approaches.